# Five-Factor Personality Dimensions Mediated the Relationship between Parents’ Parenting Style Differences and Mental Health among Medical University Students

**DOI:** 10.3390/ijerph20064908

**Published:** 2023-03-10

**Authors:** Shuxin Yao, Meixia Xu, Long Sun

**Affiliations:** 1School of Public Policy and Management (School of Emergency Management), China University of Mining and Technology, Xuzhou 221116, China; 2Department of Current Situation and Policy, School of Marxism, Shandong Women’s University, Jinan 250300, China; 3Centre for Health Management and Policy Research, School of Public Health, Cheeloo College of Medicine, Shandong University, Jinan 250012, China; 4NHC Key Laboratory of Health Economics and Policy Research (Shandong University), Jinan 250012, China

**Keywords:** five-factor personality dimensions, parenting style, mental health, medical university students

## Abstract

Background: Previous studies have identified the relationships between parental parenting style, personality, and mental health. However, the interactive influences between mother’s and father’s parenting styles on personality have been examined less often. To fill the gaps, the first aim of this study was to build the relationships between parental parenting style differences (PDs) and five-factor personality dimensions. The second aim was to test the mediating effect of five-factor personality dimensions on the relationships between parental parenting style differences and mental health. Methods: This is a cross-sectional study conducted among medical university students, and 2583 valid participants were analyzed. Mental health was measured by the Kessler-10 scale. The Chinese Big Five Personality Inventory brief version (CBF-PI-B) was used to access five-factor personality dimensions. PD was calculated by the short form of Egna Minnen av Barndoms Uppfostran. Linear regressions were conducted to analyze the associations between PD and five-factor personality dimensions. The SPSS macros program (PROCESS v3.3) was performed to test the mediating effect of five-factor personality dimensions on the associations between PD and mental health. Results: Linear regressions found that worse mental health was positively associated with PD (β = 0.15, *p* < 0.001), higher neuroticism (β = 0.61, *p* < 0.001), lower conscientiousness (β = −0.11, *p* < 0.001), lower agreeableness (β = −0.10, *p* < 0.01), and lower openness (β = −0.05, *p* < 0.05). The results also supported that PD was positively associated with lower conscientiousness (β = −0.15, *p* < 0.01), lower agreeableness (β = −0.09, *p* < 0.001), lower openness (β = −0.15, *p* < 0.001), and lower extraversion (β = −0.08, *p* < 0.001), respectively. The mediating effect of agreeableness or openness was supported for the relationships between PD and mental health. Conclusion: These findings remind us of the importance of consistent parenting styles between mother and father, and they also can be translated into practices to improve mental health among medical university students.

## 1. Introduction

Parents’ parenting style is a critical social and environmental factor for child and adolescent development. In recent years, a series of influences associated with parents’ parenting style had been identified among children and adolescents, such as cognitive and behavioral differences [1,2,3], psychological health [4,5,6], and so on [7,8,9,10]. However, most of these studies explored the respective effect of parenting style between parents; the interactive effect of parents’ parenting style was less reported.

The difference between parents’ parenting styles (PD), which can be one kind of interaction between parents, may also have an impact on children and adolescents. The reasons may be explained by the theory of cognitive dissonance [11]. As we know, parents are one of the important origins for children’s and adolescents’ cognition [12]. It implied that differing parenting styles between mothers and fathers may result in differing cognition for children and adolescents [13], and differing cognition was one of the main reasons for cognitive dissonance. Further considering the influence of cognitive dissonance on mental health [14,15], the relationship between PD and mental health can be assumed, which was also supported in our previous studies [16,17].

The five-factor model of personality was one of the most prominent models in psychometrics [18]. This model organized personality traits into five fundamental dimensions: neuroticism, extraversion, openness, agreeableness and conscientiousness [19,20]. As we know, personality traits are relatively stable across time and situation [21]. Although the building of personality was complex, parenting style was one of the important factors that contributed to the building of personality [22,23]. However, to our knowledge, no study has reported the associations between parents’ parenting style differences and the five fundamental dimensions of personality. Considering the identified impact of parents’ parenting styles on individuals’ personality, the expected relationships between PD and the five fundamental dimensions of personality may be also exist, which should be tested.

Further considering the established relationships between personality and mental health, the mediating effect of personality on the associations between PD and mental health could be built as shown in Figure 1. To fill these gaps, there were three aims for this study. The first one was to verify that mental health was associated with parenting style differences and the five fundamental dimensions of personality. The second one was to examine the associations between parents’ parenting style differences and the five fundamental dimensions of personality. The third one was to test the mediating effect of personality on the associations between parenting style differences and mental health. The findings can not only help us to further understand the associations among parenting styles, personality, and mental health but also supply fundamental information to improve adolescents’ mental health through parenting style.

## 2. Methods

### 2.1. Sample and Design

This was a cross-sectional study, which was conducted among medical university students in Shandong province, China. In total, there are 6 medical universities in Shandong province. In this study, 2 medical universities were randomly selected from these 6 medical universities. In the 2 medical universities, all the 12 majors about medicine were selected to conduct the survey. In each major, one class was randomly selected from each grade. Finally, 2723 medical university students were interviewed in this study. Because the aim of this study related to the parental parenting style differences, participants with single parents or without parents were excluded from the data analyses. After deleting the missing data about the five-factor personality and mental health, 2583 medical university students were included in this study.

### 2.2. Interview Procedure

The interview in this study was conducted in the classroom. The selected subjects were gathered in their classrooms by each class. Firstly, the interviewers introduced the aims and benefits of this study to the subjects. After signing the agreement on the written informed consent, the participants would fill in the questionnaires anonymously by themselves. Two trained interviewers would be in the classroom to answer the questions from the participants. In total, 8 interviewers participated in the interviews. All these interviewers were graduate students who majored in medicine or health administration, and they would be well trained before the interview. The interviewers were also asked to collect the questionnaires and check the questionnaires in the classroom. All the participants would not receive any credits or awards from this study. The Institutional Review Board (IRB) of Shandong University School of Public Health approved the study plan before data collection (ref. No.: 20181220).

### 2.3. Measures

#### 2.3.1. Mental Health

The Chinese version of Kessler-10 scale (K10) was used to evaluate mental health among medical university students [24,25,26]. In this scale, there were 10 items about depressive and anxiety symptoms. The participants were asked to classify the presence of these symptoms in the past four weeks. The answers for each item were from 1 (none of the time) to 5 (all the time). The sum of the scores of the 10 items represented the mental health status, and the higher scores mean the higher risk of mental health. K10 had been proved with sound validity and reliability to measure mental health in several previous studies [27,28,29], and the Chinese version of K10 was also proved with good validity and reliability [30,31].

#### 2.3.2. Five-Factor Personality

The Chinese Big Five Personality Inventory brief version (CBF-PI-B) was used to measure five-factor personality, and it was tested with nice validity and reliability among college students [32]. This scale was revised from the NEO Personality Inventory-Revised [33], and it was used by several previous studies to measure five-factor personality in China [34,35,36]. The CBF-PI-B contained 40 items to evaluate big five personality: neuroticism, conscientiousness, agreeableness, openness, and extraversion. The detailed information about this scale could be found in a previous study [32].

#### 2.3.3. Parental Parenting Style Differences (PD)

PD was calculated by the Chinese version of the short form of Egna Minnen av Barndoms Uppfostran (EMBU) among medical university students [37]. It was also used in many previous studies in China and other countries in the world [16,17,38,39]. In this scale, 21 items were applied to measure parents’ parenting styles, respectively. Three sub-scales (rejection, emotional warmth, and overprotection) were included in this scale. The total scores of PD were calculated by the following formula.
(1)$$PD=\sum_{n=i}\left|\alpha_{i}-\beta_{i}\right|$$
where *i* represents the 1st to 21st item number in the EMBU scale. $\alpha_{i}$ and $\beta_{i}$ represent the father’s and mother’s score in each item, respectively. The sum of the differences of the 21 items was analyzed in this study, and the higher scale represents the bigger parental parenting style differences.

#### 2.3.4. Social–Demographic Variables

Gender was analyzed as either male (1) or female (0). Age was calculated by medical university students’ date of birth. As there was a small percentage of other ethnicity other than Hans, ethnicity was recoded into Hans (1) and others (0). Economic status was evaluated by a self-reported question about their economic status compared with their classmates, and the answers were very good, good, average, bad, and very bad. It was recoded into good (1), average (2), and bad (3). Only children were evaluated by the question, “Are you the only child in your family?” The answers were yes (1) and no (0).

#### 2.3.5. Physical Disease

Physical disease was evaluated by one question that “Have you ever been diagnosed with any physical disease?” The answer could be chosen from yes (1) or no (0).

#### 2.3.6. Parental Relationship and Education Level

Parental relationship was evaluated by one question: “What do you think of your parents’ relationship?” The answers included very good, good, normal, bad, and very bad. As there was a small percentage for the very good and very bad selections, the parental relationship was analyzed as good (1), normal (2), and bad (3). The formal one included the very good and good selections, and the last one included recoded very bad and bad selections. The father’s and mother’s education level were evaluated by the question about their parental education level, and the answers were illiterate/semiliterate, elementary school, junior high school, junior college, bachelor, master, and doctor. As the lower report for illiterate/semiliterate, elementary school, master, and doctor, father’s and mother’s education level were analyzed as junior high school or below (1), junior college/senior school (2), and bachelor degree or above (3).

### 2.4. Statistical Analyses

IBM SPSS Statistics 24.0 (Web Edition) was used for the data analyses. Students t- test and one-way analysis of variance (ANOVA) were conducted to analyze the associated factors of mental health for categorical variables. Correlation analyses were conducted to analyze correlations between continuous variables and K10 scores. Linear regressions were also conducted to analyze the factors associated with mental health. To test the mediating effect of five-factor personality dimensions, the SPSS macros program (PROCESS v3.3) developed by Andrew F. Hayes was used [40]. All the tests were two-tailed, and a *p*-value of ≤ 0.05 was considered statistically significant.

## 3. Results

In this study, 2583 medical university students were analyzed. The sample characteristics are described in Table 1. The mean age for these students was 20.22 years, with a standard deviation of 1.27. The other detailed information could be found in the second column in Table 1. Results about the single analyses for the factors related to mental health are also shown in this table. The results supported that worse mental health was positively associated with older age (r = 0.104, *p* < 0.001), other ethnicity (t = −2.116, *p* < 0.05), physical disease (t = 6.034, *p* < 0.001), worse economic status (F = 11.829, *p* < 0.001), and worse parental relationship (F = 18.167, *p* < 0.001). The other detailed information can be found in Table 1.

In Table 2, the correlation coefficient matrix for PD, five-factor personality dimensions, and mental health was analyzed. The results supported that worse mental health was positively associated with higher neuroticism (r = 0.498, *p* < 0.001), lower conscientiousness (r = −0.069, *p* < 0.001), and lower openness (r = −0.079, *p* < 0.001). However, mental health was not supported to be associated with agreeableness (r = 0.027, *p* > 0.05) and extraversion (r = 0.002, *p* > 0.05). The associations between parental parenting styles differences and mental health were also supported (r = 0.146, *p* < 0.001) in this study. The detailed information for the bivariate correlations are shown in Table 2.

Linear regressions were also conducted to analyze the factors associated with mental health among medical university students. In model 1, the factors associated with worse mental health were older age (β = 0.48, *p* < 0.001), not Hans ethnicity (β = −1.49, *p* < 0.05), physical disease (β = 3.37, *p* < 0.001), good economic status (β = −0.96, *p* < 0.01), good parental relationship (β = −1.38, *p* < 0.001), and PD (β = 0.15, *p* < 0.001). In Model 2, we further added five-factor personality dimensions into the regression, and the results showed that worse mental health was positively associated with male (β = 0.73, *p* < 0.01), older age (β = 0.45, *p* < 0.001), physical disease (β = 2.34, *p* < 0.001), bad parental relationship (β = −0.37, *p* < 0.05), larger PD (β = 0.15, *p* < 0.001), higher neuroticism (β = 0.61, *p* < 0.001), lower conscientiousness (β = −0.11, *p* < 0.001), lower agreeableness (β = −0.10, *p* < 0.01), and lower openness (β = −0.05, *p* < 0.05). The detailed information is shown in Table 3.

To build the mediating effect of five-factor personality dimensions, linear regressions were also conducted to analyze the association between PD and five-factor personality dimensions. The results showed that larger PD was associated with lower conscientiousness (β = −0.15, *p* < 0.01), lower agreeableness (β = −0.09, *p* < 0.001), lower openness (β = −0.15, *p* < 0.001), and lower extraversion (β = −0.08, *p* < 0.001), respectively. The detailed results for the linear regression are listed in Table 4.

The mediating effect of five-factor dimensions on the association between PD and mental health is shown in Figure 2. As the results did not support the associations between extraversion and mental health (β = −0.02, *p* > 0.05 in Table 3), nor the association between PD and neuroticism (β = 0.02, *p* > 0.05 in Table 4), the mediating effects of conscientiousness, agreeableness, and openness were explored in this study. The results supported that agreeableness and openness could, respectively, mediate the association between PD and mental health. The detailed information can be found in Figure 2.

## 4. Discussion

There were several critical findings for this study. The first one was that worse mental health was positively associated with larger PD, higher neuroticism, lower conscientiousness, lower agreeableness, and lower openness. The second one was that larger PD was associated with conscientiousness, agreeableness, openness, and extraversion. The final finding was that conscientiousness or agreeableness could mediate the relationships between PD and mental health.

In this study, the first finding was about the association between mental health, PD, and personality. The results supported that mental health was associated with PD, neuroticism, conscientiousness, agreeableness, and openness. These associations were not unexpected. The association between PD and mental health is supported by previous studies [16,17]. The reason can be explained by the theory of cognitive dissonance [11], as we explained in the Introduction section. Indeed, these associations were also supported by previous studies [41,42]. However, extraversion was not associated with mental health. A previous longitudinal study found that extraversion played roles on mental health through neuroticism [43]. After controlling neuroticism, the association between extraversion and mental health may disappear.

One of the main aims for this study was to test the associations between PD and five-factor personality dimensions. The results supported that PD was negatively associated with conscientiousness, agreeableness, openness, and extraversion. Indeed, the associations between personality and PD was not surprising. The associations between parenting style and personality had been identified in many previous studies [5,44,45]. The reasons could be explained by the theory of cognitive dissonance [46], as we introduced in the Introduction section. As we know, the parenting style was associated with their father and mother. Because of this, we can further suppose the different parenting styles between mother and father may be associated with personality. The identified associations between parenting styles and borderline personality disorder also could imply the associations [47].

Our results did not support the association between PD and neuroticism, although previous studies had found that mental health was associated with both PD and neuroticism [16,48,49]. As we know, the building of neuroticism traits was a gradual process. Parenting styles mainly play more roles on neuroticism in the early stage of life [50,51], and the association between parenting style and neuroticism would be weaker in later life. Although neuroticism is characterized by psychological problems [19], the associations between neuroticism and mental health may strengthen in later life. A longitudinal study found that higher neuroticism was associated with more negative daily experiences [52], and another study also supported that negative daily experiences were associated with mental health [53]. As we know, negative daily experiences happen in everyone’s lives. The more one lives, the more negative daily experiences one has. This may further explain why neuroticism was associated with mental health, but PD was not associated with the neuroticism trait.

The other important aim was to explore the mediating effect of five-factor personality dimensions on the association between PD and mental health. Based on the previous findings in this study, the mediating effects of conscientiousness, agreeableness, and openness were tested, and the results supported that agreeableness and openness could mediate the association between PD and mental health. Considering the stability of personality across time and situation to people [21], we can further assume the stable effect of PD on agreeableness, openness, and mental health.

This study also supported that worse mental health was positively related to being male, older age, physical disease, and bad parental relationship. As all these factors had been identified in many previous studies [54,55]; these factors were not discussed here. This study also supported that males scored lower in all the five-factor personality dimensions than females. Previous studies had proved that girls matured more quickly than boys in the various personality dimensions [56]. The age differences in five-factor personality dimensions were also reported in previous studies [57]. The relationships between parental relationship and family environment were also supported in previous study [58].

Some limitations should be considered when these findings are interpreted. Firstly, as a cross-sectional design, the causal relationships among these factors should be cautious. Secondly, all of variables were self-reported, and the recall bias cannot be avoided in this study. Finally, the data were from the medical university students, and the consistency of our findings in other populations should be tested.

Keeping these limitations in mind, there were several critical findings in this study. Firstly, our results further supported that mental health was associated with PD, neuroticism, conscientiousness, agreeableness, and openness. Secondly, PD was associated with conscientiousness, agreeableness, openness, and extraversion. Finally, the mediating effect of agreeableness or openness on the associations between PD and mental health was also supported in this study. These findings remind us of the importance of consistent parenting styles between mother and father, and they also can be translated into practices to improve mental health among medical university students. The further studies could test the mediating effect of personality on the relationships between PD and mental health based on a longitudinal design.

## Figures and Tables

**Figure 1 ijerph-20-04908-f001:**
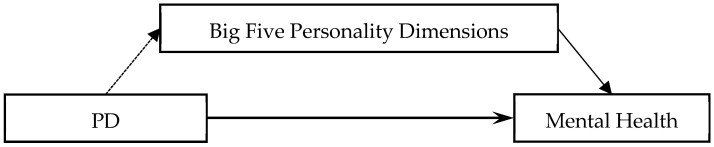
Hypothesis model. Note: PD denotes total differences for parental parenting styles.

**Figure 2 ijerph-20-04908-f002:**
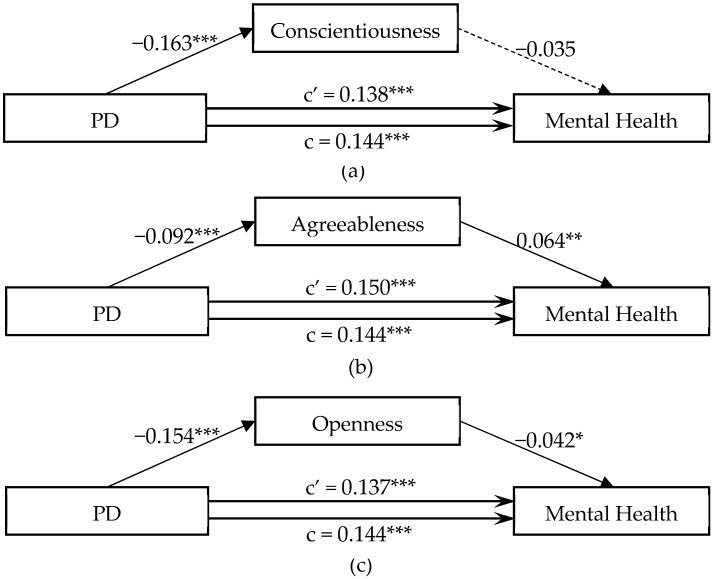
Mediating effect of big five personality dimensions on the associations between PD and mental health. Note: PD denotes total differences for parental parenting styles. * refers to *p* < 0.05; ** refers to *p* < 0.01; *** refers to *p* < 0.001.

**Table 1 ijerph-20-04908-t001:** Sample description and single analyses for the factors associated with mental health.

Variables	Mean ± SD/n (%)	Mental Health (Mean ± SD)	t/r/F
Total	2583 (100.0)	19.14 ± 6.65	--
Gender			t = 0.961
Male	1151 (44.6)	19.28 ± 6.98	
Female	1432 (55.4)	19.03 ± 6.37	
Age	20.22 ± 1.27	19.14 ± 6.65	r = 0.104 ***
Ethnicity			t = −2.116 *
Hans	2504 (95.9)	19.09 ± 6.61	
Others	79 (3.1)	20.70 ± 7.69	
Physical disease			t = 6.034 ***
Yes	113 (4.4)	22.81 ± 7.06	
No	2470 (95.6)	18.97 ± 6.58	
Economic status			F = 11.829 ***
Good	520 (20.1)	18.18 ± 6.81	
Average	1786 (69.1)	19.20 ± 6.54	
Bad	277 (10.7)	20.56 ± 6.78	
Only child			t = −1.270
Yes	1190 (46.1)	18.96 ± 6.88	
No	1393 (53.9)	19.29 ± 6.44	
Parental relationship			F = 18.167 ***
Good	2170 (84.0)	18.80 ± 6.51	
Normal	336 (13.0)	20.79 ± 6.83	
Bad	77 (3.0)	21.48 ± 8.10	
Father’s education			F = 2.238
Junior high school or below	1312 (50.8)	19.41 ± 6.32	
Junior college/senior school	887 (34.3)	18.85 ± 6.86	
Bachelor degree or above	384 (14.9)	18.88 ± 7.20	
Mother’s education			F = 2.171
Junior high school or below	1562 (60.5)	19.33 ± 6.52	
Junior college/senior school	720 (27.9)	18.98 ± 6.70	
Bachelor degree or above	301 (11.7)	18.52 ± 7.14	

Note: SD denotes standard deviation. * refers to *p* < 0.05; *** refers to *p* < 0.001. The values of t are calculated from Student’s t test. The values of F are calculated from analysis of variance. The values of r are calculated from bivariate correlation.

**Table 2 ijerph-20-04908-t002:** Correlation coefficient matrix for parenting style differences, big five personality dimensions, and mental health (n = 2583).

	Mean ± SD	1.1	1.2	1.3	1.4	1.5	2
1. Big five personality dimensions							
1.1 Neuroticism	23.65 ± 6.41	1					
1.2 Conscientiousness	29.28 ± 6.85	0.308 ***	1				
1.3 Agreeableness	28.09 ± 5.80	0.444 ***	0.668 ***	1			
1.4 Openness	30.10 ± 7.47	0.261 ***	0.706 ***	0.648 ***	1		
1.5 Extraversion	27.63 ± 6.11	0.368 ***	0.679 ***	0.627 ***	0.763 ***	1	
2. Mental health	19.14 ± 6.65	0.498 ***	−0.069 ***	0.027	−0.079 ***	0.002	1
3. PD	4.33 ± 5.97	0.013	−0.160 ***	−0.116 ***	−0.134 ***	−0.090 ***	0.146 ***

Note: PD denotes total differences for parental parenting styles. SD denotes standard deviation. *** refers to *p* < 0.001.

**Table 3 ijerph-20-04908-t003:** Linear regression for the factors associated with mental health (β (95% CI)).

	Model 1	Model 2
n	2583	2583
Male	0.31 (−0.22, 0.84)	0.73 (0.29, 1.18) **
Age	0.48 (0.29, 0.68) ***	0.45 (0.28, 0.62) ***
Hans ethnicity	−1.49 (−2.94, −0.03) *	−1.08 (−2.30, 0.14)
Physical disease	3.37 (2.14, 4.60) ***	2.34 (1.30, 3.37) ***
Economic status (Ref. = Average)		
Good	−0.96 (−1.63, −0.29) **	−0.56 (−1.12, 0.01)
Bad	0.42 (−0.43, 1.27)	0.10 (−0.62, 0.81)
Only child	−0.46 (−1.03, 0.10)	−0.42 (−0.89, 0.05)
Parental relationship (Ref. = Normal)		
Good	−1.38 (−2.14, −0.61) ***	−0.73 (−1.37, −0.08) *
Bad	0.30 (−1.32, 1.91)	−0.47 (−1.83, 0.89)
Father’s education (Ref. = Junior high school or below)		
Junior college/senior school	−0.23 (−0.86, 0.40)	−0.07 (−0.60, 0.46)
Bachelor degree or above	0.32 (−0.72, 1.36)	0.51 (−0.36, 1.38)
Mother’s education (Ref. = Junior high school or below)		
Junior college/senior school	0.15 (−0.53, 0.82)	0.26 (−0.31, 0.83)
Bachelor degree or above	−0.39 (−1.51, 0.73)	−0.38 (−1.32, 0.55)
PD	0.15 (0.10, 0.19) ***	0.10 (0.06, 0.14) ***
Neuroticism	--	0.61 (0.57, 0.65) ***
Conscientiousness	--	−0.11 (−0.16, −0.06) ***
Agreeableness	--	−0.10 (−0.15, −0.04) **
Openness	--	−0.05 (−0.10, −0.002) *
Extraversion	--	−0.02 (−0.08, 0.04)
Constant	11.41 (7.08, 15.75) ***	4.78 (0.90, 8.66) *
R^2^	0.060	0.343

Note: CI denotes the confidence interval. PD denotes total differences for parental parenting styles. * refers to *p* < 0.05; ** refers to *p* < 0.01; *** refers to *p* < 0.001.

**Table 4 ijerph-20-04908-t004:** Linear regression for the factors associated with big five personality dimensions (β (95% CI)).

	Neuroticism	Conscientiousness	Agreeableness	Openness	Extraversion
n	2583	2583	2583	2583	2583
Male	−1.15 (−1.67, −0.63) ***	−0.85 (−1.40, −0.30) **	−1.34 (−1.80, −0.88) ***	−0.80 (−1.40, −0.20) **	−0.69 (−1.19, −0.20) **
Age	−0.06 (−0.26, 0.13)	−0.15 (−0.36, 0.06)	−0.28 (−0.45, −0.10) **	−0.45 (−0.68, −0.23) ***	−0.29 (−0.47, −0.10) **
Hans ethnicity	−0.28 (−1.71, 1.16)	1.08 (−0.43, 2.60)	0.87 (−0.42, 2.15)	0.62 (−1.04, 2.28)	0.11 (−1.25, 1.48)
Physical disease	1.57 (0.37, 2.78) **	−0.47 (−1.75, 0.80)	−0.62 (−1.70, 0.46)	0.68 (−0.71, 2.08)	0.11 (−1.04, 1.26)
Economic status (Ref. = Average)					
Good	−1.09 (−1.75, −0.43) **	−0.92 (−1.61, −0.22) **	−1.19 (−1.78, −0.60) ***	−0.69 (−1.45, 0.07)	−0.38 (−1.01, 0.25)
Bad	0.26 (−0.58, 1.09)	−0.78 (−1.67, 0.10)	−0.40 (−1.15, 0.35)	−0.56 (−1.53, 0.41)	−0.48 (−1.28, 0.32)
Only child	−0.22 (−0.78, 0.33)	−0.62 (−1.21, −0.03) *	−0.23 (−0.73, 0.27)	−0.03 (−0.67, 0.61)	−0.08 (−0.61, 0.45)
Parental relationship (Ref. = Normal)					
Good	−0.53 (−1.28, 0.22)	1.60 (0.80, 2.39) ***	0.80 (0.12, 1.47) *	0.99 (0.12, 1.87) *	0.98 (0.26, 1.70) **
Bad	1.60 (0.01, 3.19) *	1.41 (−0.27, 3.09)	0.51 (−0.91, 1.94)	−0.09 (−1.92, 1.75)	0.44 (−1.08, 1.95)
Father’s education (Ref. = Junior high school or below)					
Junior college/senior school	−0.16 (−0.79, 0.46)	0.24 (−0.42, 0.89)	0.02 (−0.53, 0.58)	0.61 (−0.11, 1.33)	0.18 (−0.42, 0.77)
Bachelor degree or above	−0.28 (−1.30, 0.75)	0.04 (−1.04, 1.12)	0.05 (−0.86, 0.97)	0.25 (−0.93, 1.43)	−0.05 (−1.02, 0.92)
Mother’s education (Ref. = Junior high school or below)					
Junior college/senior school	−0.04 (−0.70, 0.63)	0.29 (−0.42, 0.99)	0.11 (−0.49, 0.71)	0.69 (−0.09, 1.46)	0.62 (−0.02, 1.25)
Bachelor degree or above	0.20 (−0.90, 1.29)	0.56 (−0.60, 1.72)	0.06 (−0.93, 1.04)	0.96 (−0.31, 2.22)	0.54 (−0.51, 1.59)
PD	0.02 (−0.02, 0.06)	−0.15 (−0.20, −0.11) ***	−0.09 (−0.12, −0.05) ***	−0.15 (−0.20, −0.10) ***	−0.08 (−0.12, −0.04) ***
Constant	26.32 (22.06, 30.58) ***	31.22 (26.72, 35.73) ***	33.51 (29.69, 37.32) ***	38.49 (33.57, 43.42) ***	32.96 (28.90, 37.02) ***
R^2^	0.025	0.044	0.041	0.037	0.022

Note: CI denotes to confidence interval. PD denotes total differences for parental parenting styles. * refers to *p* < 0.05; ** refers to *p* < 0.01; *** refers to *p* < 0.001.

## Data Availability

The datasets used and/or analyzed during the current study are available from the corresponding author on reasonable request.

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
