# Peer review of "Five-Factor Personality Dimensions Mediated the Relationship between Parents’ Parenting Style Differences and Mental Health among Medical University Students"

_ijerph, 2023, doi:10.3390/ijerph20064908_

Round 1

Reviewer 1 Report

The subject and aim of this research is original and interesting.

The introduction provides pertinent and relevant information to understand the research problem and the aims of the researchers.

The methodology is accurately described in detail and is coherent with the stablished goals.

The abstract needs to be corrected because it’s unclear regarding statistical analyses and results. I suggest redacting conjointly the specific analyses with the main results: linear regressions by order, and then, the mediation model.

In general, the English writing needs to be reviewed to enhance the quality of the article.

Tables need to be improved, especially regarding the explicative foot note.

Table 1: The notation t/r/F is no explained at the foot note. Also, at the values notations it’s needed to add spaces between symbols and numbers, by example:

19.14+6.65 should be written as 19.14 + 6.65; t=0.961 should be t = 0.961 and so on.,

The same applies in the text when reporting results, and on others tables with symbols/number notations.

Table 2 lacks spaces too. I suggest aligning the columns of variables to the left side.

Notation of p values will be clearer if the authors use: or =  to connect * to p value.

Table 3. Betas and confidence interval should be indicated at the top of the table under Model 1 and Model 2 notation.

The results are congruent with the methodology and the goals. They are very interesting and in consequence it should be discussed in order and more exhaustively.

By example the use of the K10 scale to evaluate mental health should be explained and highlighted, given the fact that the scale measures self-reported psychological distress, meaning that higher values indicate worse mental health. That’s the reason that in the linear regression mental health and PD association is positive instead of negative, that could be the result we expected.

Also, table 4 gives us very interesting information about differences regarding gender, ethnicity, economic status parental relationship etc. that should be discussed to a better understanding of the diversity of factors influencing the personality traits model under study.

Author Response

Reviewer 1

The subject and aim of this research is original and interesting. The introduction provides pertinent and relevant information to understand the research problem and the aims of the researchers. The methodology is accurately described in detail and is coherent with the stablished goals.

[Response] Thanks for your nice comments on our manuscript. We have read your comments carefully, and all of them are very helpful for us to revise this manuscript. We have tried our best to revise this manuscript. We wish we have fully understood your comments, and given reasonable responses to these comments. Following are the point-by-point responses. Please feel free to let us know if you have any other comments on our manuscript. Thanks for your time and effort on our manuscript.

The abstract needs to be corrected because it’s unclear regarding statistical analyses and results. I suggest redacting conjointly the specific analyses with the main results: linear regressions by order, and then, the mediation model.

[Response] Thanks for your suggestion. We have revised the methods and results section in the Abstract according to your suggestions. In this version, we have redacted conjointly the main results by the order you suggested. Following are the revised paragraphs.

Methods: This is a cross-sectional study conducted among medical university students, and 2583 valid participants were analyzed. Mental health was measured by Kessler-10 scale. The Chinese Big Five Personality Inventory brief version (CBF-PI-B) was used to access five-factor personality dimensions. PD was calculated by the short form of Egna Minnen av Barndoms Uppfostran. Linear regressions were conducted to analyze the associations between PD and five-factor personality dimensions. SPSS macros program (PROCESS v3.3) was performed to test the mediating effect of five-factor personality dimensions on the associations between PD and mental health. Results: Linear regressions found that worse mental health was positively associated with PD (β = 0.15, p < 0.001), higher neuroticism (β = 0.61, p < 0.001), lower conscientiousness (β = -0.11, p < 0.001), lower agreeableness (β = -0.10, p < 0.01), and lower openness (β = -0.05, p < 0.05). The results also supported that PD was positively associated with lower conscientiousness (β = -0.15, p < 0.01), lower agreeableness (β = -0.09, p < 0.001), lower openness (β = -0.15, p < 0.001), and lower extraversion (β = -0.08, p < 0.001), respectively. The mediating effect of agreeableness or openness was supported on the relationships between PD and mental health.

In general, the English writing needs to be reviewed to enhance the quality of the article.

[Response] Thanks for your suggestions about the language problems. We are so sorry about it. As you may know, English is not our first language, and it is really hard for us to avoid all the language problems. However, we have read this manuscript throughout, and tried our best to revise this manuscript. We think most of the language problems had been revised. As these were too many revisions, we cannot copy the revisions here, and please check the revisions in the revised manuscript. Thanks for your suggestions.

Tables need to be improved, especially regarding the explicative foot note.

[Response] Thanks for your reminding. We have checked all the footnotes in the Tables. As some suggestions were listed in the following questions, we will response this question in the following questions. Thanks for your suggestion.

Table 1: The notation t/r/F is no explained at the foot note. Also, at the values notations it’s needed to add spaces between symbols and numbers, by example: 19.14+6.65 should be written as 19.14 + 6.65; t=0.961 should be t = 0.961 and so on.

[Response] Thanks for your suggestion. We have explained t/r/F in the footnote. The added sentences were “The values of t are calculated from Student’s t test. The values of F are calculated from analysis of variance. The values of r are calculated from bivariate correlation.” We also add spaces between symbols and numbers throughout this manuscript. Thanks for your suggestion.

The same applies in the text when reporting results, and on others tables with symbols/number notations.

[Response] Thanks for your reminding. We have revised the text and tables throughout this manuscript according to your suggestion.

Table 2 lacks spaces too. I suggest aligning the columns of variables to the left side.

[Response] Similar to the last question. Table 2 was also revised according to your suggestion. We also make the tables aligning the columns of variables to the left side.

Notation of p values will be clearer if the authors use: or =to connect * to p value.

[Response] We are so sorry about this question. We are not sure if we have fully understood your questions. In this revision, we have revised the footnotes about * in to the following words: “* refers to p < 0.05; ** refers to p < 0.01; *** refers to p < 0.001.

Table 3. Betas and confidence interval should be indicated at the top of the table under Model 1 and Model 2 notation.

[Response] Thanks for your suggestion. We have listed the betas and confidence interval at the top of the Table 3 and Table 4.

The results are congruent with the methodology and the goals. They are very interesting and in consequence it should be discussed in order and more exhaustively.

[Response] Thanks for your suggestions. We have revised the whole Discussion section. As all the Discussion section has been revised, we do not list the revision here. Please check the revisions in the manuscript. Sorry about it.

By example the use of the K10 scale to evaluate mental health should be explained and highlighted, given the fact that the scale measures self-reported psychological distress, meaning that higher values indicate worse mental health. That’s the reason that in the linear regression mental health and PD association is positive instead of negative, that could be the result we expected.

[Response] Thank you so much for your reminding. We do agree with you. Mental health is a neutral word. However, the higher scale scores of K10 means the worse mental health in this study. Because of this, we should add worse before mental health throughout this manuscript in this study. We have checked the whole manuscript, and revised all the words about this problem. Thank you so much for your reminding about this problem.

Also, table 4 gives us very interesting information about differences regarding gender, ethnicity, economic status parental relationship etc. that should be discussed to a better understanding of the diversity of factors influencing the personality traits model under study.

[Response] Thanks for your suggestion. We have added a paragraph to discuss the associations between social-demographic variables and personality. Following are the added paragraph.

This study also supported that worse mental health was positively related to male, older age, physical disease, and bad parental relationship. As all these factors had been identified in many previous studies [54, 55], these factors were not discussed here. This study also supported that males were in lower level in all the five-factor personality dimensions than females. Previous studies had proved that girls matured more quickly than boys in the various personality dimensions [56]. The age differences in five-factor personality dimensions were also reported in previous studies [57]. For the relationships between parental relationship and family environment were also supported in previous study [58].

Finally, we should thank you for your time and effort on our manuscript. In this revision, we have tried our best to revise this manuscript. However, we do not know if we have fully understood your comments. Please feel free to let us know if you have any other questions.

Reviewer 2 Report

Parents parenting style is a critical social and environmental factor for child and adolescent development. So far,  most studies explored mother’s and father’s parenting style separately, and the interactive effect of parents’ parenting style was less reported. Thus,   the first aim of the study was to build  the associations between parental parenting style differences (PD) and five-factor personality dimensions. The second aim was to test the mediating effect of five-factor personality dimensions on the associations between parental parenting style differences and mental health.

The paper has a clear  structure (Introduction, Methods, Results and   Discussion)) and the subject seems to be  interesting and useful, however it is not so easy to follow due to the many specialized terms and scales that are not always fully explained. The manuscript stands out for its detailed methodology and good in-depth analysis of the results; discussion looks sufficient and contain  the limitation section. The text is complemented by  two transparent figures, four tables and  enriched with 53 adequate references.

 However there are a few  issues which could be improved:

1.     In Methods , subsection Mental health (line 105-106)  there is a sentence:

‘Sum of the scores of the 10 items represented the mental health status, and the higher scores mean the higher risk of mental health’. It sounds vague: higher risk of good or bad mental health status?

2.     In Results the following sentence  (line 169-171) seems to be incomplete and needs to be corrected: ‘The results supported that age (r=0.104, p<0.001), ethnicity (t=-2.116,

p<0.05), physical disease (t=6.034, p<0.001), economic status (F=11.829, p<0.001), and parental relationship (F=18.167, p<0.001).’

3.     In Results the following sentence  (line 189-193) looks vague and needs to be explained and rebuilt: ‘In Model 2, we further added five-factor personality 189 dimensions into the regression, and the results showed that mental health was male 190  (β=0.73, p<0.01), age (β=0.45, p<0.001), physical disease (β=2.34, p<0.001), good parental  relationship (β=-0.37, p<0.05), PD (β=0.15, p<0.001), neuroticism (β=0.61, p<0.001), conscientiousness (β=-0.11, p<0.001), agreeableness (β=-0.10, p<0.01), openness (β=-0.05, p<0.05).’

How mental health is related to male, age and other factors?

4.     In Table 1, the statistical abbreviations "t", "r" and "F" should be explained in the legend.

5.     In Discussion the following sentence sounds  vague and needs to be clarified: ] . ‘Further considering the identified associations between negative daily experiences and mental health [53], we may explain why neuroticism was associated with mental health, but PD was not associated with neuroticism trait.’ (line 248-250). How we can explain it?

6.     The conclusion of the study is very brief and should be extended with practical implications for the future. In which way the findings of the study could improve mental health among medical college  students ? Be more specific.

7. It would be worthwhile to attach to the manuscript all used scales and tests in the study in the form of Appendix.

Author Response

Reviewer 2

Parents parenting style is a critical social and environmental factor for child and adolescent development. So far, most studies explored mother’s and father’s parenting style separately, and the interactive effect of parents’ parenting style was less reported. Thus, the first aim of the study was to build the associations between parental parenting style differences (PD) and five-factor personality dimensions. The second aim was to test the mediating effect of five-factor personality dimensions on the associations between parental parenting style differences and mental health.

The paper has a clear structure (Introduction, Methods, Results and Discussion)) and the subject seems to be interesting and useful, however it is not so easy to follow due to the many specialized terms and scales that are not always fully explained. The manuscript stands out for its detailed methodology and good in-depth analysis of the results; discussion looks sufficient and contain the limitation section. The text is complemented by two transparent figures, four tables and enriched with 53 adequate references.

[Response] Thanks for your nice comments on our manuscript. We have read your comments carefully, and all of them are very helpful for us to revise this manuscript. We have tried our best to revise this manuscript. We wish we have fully understood your comments, and given reasonable responses to these comments. Following are the point-by-point responses. Please feel free to let us know if you have any other comments on our manuscript. Thanks for your time and effort on our manuscript.

However, there are a few issues which could be improved:

  1. In Methods, subsection Mental health (line 105-106) there is a sentence:

‘Sum of the scores of the 10 items represented the mental health status, and the higher scores mean the higher risk of mental health’. It sounds vague: higher risk of good or bad mental health status?

[Response] Thank you so much for your reminding. We do agree with you. Mental health is a neutral word. However, the higher scale scores of K10 means the worse mental health in this study. Because of this, we should add worse before mental health throughout this manuscript in this study. We have checked the whole manuscript, and revised all the words about this problem. Thank you so much for your reminding about this problem.

  1. In Results the following sentence (line 169-171) seems to be incomplete and needs to be corrected: ‘The results supported that age (r=0.104, p<0.001), ethnicity (t=-2.116, p<0.05), physical disease (t=6.034, p<0.001), economic status (F=11.829, p<0.001), and parental relationship (F=18.167, p<0.001).’

[Response] We are so sorry about this problem. It is really our careless. We have revised this sentence, and the revised sentence is “The results supported that worse mental health was positively associated with older age (r = 0.104, p < 0.001), other ethnicity (t = -2.116, p < 0.05), physical disease (t = 6.034, p < 0.001), worse economic status (F = 11.829, p < 0.001), and worse parental relationship (F = 18.167, p < 0.001).”

  1. In Results the following sentence (line 189-193) looks vague and needs to be explained and rebuilt: ‘In Model 2, we further added five-factor personality 189 dimensions into the regression, and the results showed that mental health was male 190 (β=0.73, p<0.01), age (β=0.45, p<0.001), physical disease (β=2.34, p<0.001), good parental relationship (β=-0.37, p<0.05), PD (β=0.15, p<0.001), neuroticism (β=0.61, p<0.001), conscientiousness (β=-0.11, p<0.001), agreeableness (β=-0.10, p<0.01), openness (β=-0.05, p<0.05).’ How mental health is related to male, age and other factors?

[Response] Similar to the last question. It is really our problems. We are so sorry about it. We have revised this sentence, and the revised sentence is “In Model 2, we further added five-factor personality dimensions into the regression, and the results showed that worse mental health was positively associated with male (β = 0.73, p < 0.01), older age (β = 0.45, p < 0.001), physical disease (β = 2.34, p < 0.001), bad parental relationship (β = -0.37, p < 0.05), larger PD (β = 0.15, p < 0.001), higher neuroticism (β = 0.61, p < 0.001), lower conscientiousness (β = -0.11, p < 0.001), lower agreeableness (β = -0.10, p < 0.01), and lower openness (β = -0.05, p < 0.05).”

  1. In Table 1, the statistical abbreviations "t", "r" and "F" should be explained in the legend.

[Response] Thanks for your reminding. We have listed the detailed information about t, r, F in the footnotes of Table 1. The added sentences are “The values of t are calculated from Student’s t test. The values of F are calculated from analysis of variance. The values of r are calculated from bivariate correlation.”

  1. In Discussion the following sentence sounds vague and needs to be clarified:]. ‘Further considering the identified associations between negative daily experiences and mental health [53], we may explain why neuroticism was associated with mental health, but PD was not associated with neuroticism trait.’ (line 248-250). How we can explain it?

[Response] Thanks for your suggestions. We have revised these sentences as follows “A longitudinal study found that higher neuroticism was associated with more negative daily experiences [52], and other study also supported that negative daily experiences was associated with mental health [53]. As we know, negative daily experiences would happen in all the lives. The more life, the more negative daily experiences. This may further explain why neuroticism was associated with mental health, but PD was not associated with neuroticism trait.”

  1. The conclusion of the study is very brief and should be extended with practical implications for the future. In which way the findings of the study could improve mental health among medical college students? Be more specific.

[Response] Thanks for your suggestions. We have added sentences to give suggestions on the further study and practical implications. The revised sentences were “These findings remind us the importance of consistent parenting styles between mother and father, and they also can be translated into practices to improve mental health among medical university students. The further studies could test the mediating effect of personality on the relationships between PD and mental health based on a longitudinal design.

  1. It would be worthwhile to attach to the manuscript all used scales and tests in the study in the form of Appendix.

[Response] Thanks for your suggestion. We do agree with you, attaching all the used scales and tests in the Appendix were very important for the readers to follow this study. However, as you may know, we did not find how to submit the Appendix. In the other sides, as you know, all the scales used in this study was in Chinese, and it may be not interesting for the international readers. Thanks for your suggestions on our manuscript.

Finally, we should thank you for your time and effort on our manuscript. In this revision, we have tried our best to revise this manuscript. However, we do not know if we have fully understood your comments. Please feel free to let us know if you have any other questions.

Round 2

Reviewer 1 Report

Congratulations for the improvement in your paper! I'm glad to know that my comments have been useful to you.